# Phase IIa Clinical Biomarker Trial of Dietary Arginine Restriction and Aspirin in Colorectal Cancer Patients

**DOI:** 10.3390/cancers15072103

**Published:** 2023-03-31

**Authors:** Jason A. Zell, Thomas H. Taylor, C. Gregory Albers, Joseph C. Carmichael, Christine E. McLaren, Lari Wenzel, Michael J. Stamos

**Affiliations:** 1Division of Hematology/Oncology, Department of Medicine, University of California Irvine Medical Center, Orange, CA 92868, USA; 2Chao Family Comprehensive Cancer Center, University of California Irvine Medical Center, Orange, CA 92868, USA; 3Department of Epidemiology & Biostatistics, University of California Irvine, Irvine, CA 92697, USA; 4Division of Gastroenterology, Department of Medicine, University of California Irvine Medical Center, Orange, CA 92868, USA; 5Division of Colorectal Surgery, Department of Surgery, University of California Irvine Medical Center, Orange, CA 92868, USA; 6Department of Medicine, University of California Irvine Medical Center, Orange, CA 92868, USA

**Keywords:** arginine restriction, aspirin, clinical trial, colorectal cancer, cancer prevention, dietary intervention, polyamines

## Abstract

**Simple Summary:**

The authors present a phase IIa clinical biomarker trial of colorectal cancer (CRC) survivors treated with a polyamine-inhibitory regimen (dietary arginine restriction plus daily aspirin 325 mg) as a potential future strategy for tertiary prevention of CRC (i.e., preventing colorectal adenomas and/or CRC among CRC survivors). In this study, 20 stage I-III CRC patients were treated for 12 weeks with the above-mentioned intervention, with assessments of calculated dietary arginine intake, plasma arginine levels, and rectal tissue mucosa samples assessed for polyamine levels at baseline and end-of-study. While dietary arginine intake and plasma arginine levels were significantly decreased by this intervention, rectal tissue polyamines (including tissue putrescine levels, which served as the primary study endpoint) were unaffected.

**Abstract:**

After potentially curative treatment, colorectal cancer (CRC) patients remain at high risk for recurrence, second primary CRC, and high-risk adenomas. In combination with existing data, our previous findings provide a rationale for reducing tissue polyamines as tertiary prevention in non-metastatic CRC patients. The goal of this study was to demonstrate rectal tissue polyamine reduction in optimally treated stage I-III CRC patients after intervention with daily oral aspirin + dietary arginine restriction. A single-institution phase IIa clinical trial was conducted. Patients were treated with aspirin 325 mg/day and an individualized dietary regimen designed to reduce arginine intake by ≥30% over a 12-week study period. Dietary intake, endoscopy with rectal biopsies, and phlebotomy were performed pre- and post-intervention. The primary endpoint was to demonstrate ≥50% decrease in rectal tissue putrescine levels from baseline as a measure of polyamine reduction in the target tissue. Twenty eligible patients completed the study. After study intervention, mean dietary arginine intake decreased from 3.7 g/day ± 1.3 SD to 2.6 g/day ± 1.2 SD (29.7% decrease, *p* < 0.02 by Sign test). Mean plasma arginine levels decreased from 46.0 ng/mL ± 31.5 SD at baseline to 35 ng/mL ± 21.7 SD (*p* < 0.001). Rectal tissue putrescine levels were 0.90 nMol/mg-protein pre-intervention and 0.99 nMol/mg-protein post-intervention (*p* < 0.64, NS). No significant differences were observed for the other tissue polyamines investigated: spermidine (*p* < 0.13), spermine (*p* < 0.21), spermidine:spermine ratio (*p* < 0.71). Among CRC survivors, treatment with daily oral aspirin and an individualized dietary arginine restriction intervention resulted in lower calculated dietary arginine intake and plasma arginine levels but did not affect rectal tissue polyamine levels.

## 1. Introduction

An estimated 106,970 new cases of colon cancer and 46,050 cases of rectal cancer will occur in the U.S. in 2023, making colorectal cancer (CRC) the third most common cancer among U.S. males and females [1]. Worldwide, CRC ranks as the 3rd most common form of cancer, with 1.9 million cases in 2020 (the most recent year for which we have global data); it ranks as the 2nd leading global cancer cause of death (916,000 deaths) [2]. Despite cure rates exceeding 90% for localized (stage I) disease, adherence to CRC screening is low (50–60%) and therefore 2/3rds of patients are diagnosed with regional (stage II, III) or advanced (stage IV) disease. As such, relative survival is estimated at just 65% [1]. Aside from risk of recurrence and CRC-specific mortality, CRC patients remain at high risk for multiple competing events. Such events include a 50% risk of recurrent adenoma and 27% risk of either a second primary colorectal cancer or high-risk adenoma (i.e., high-grade dysplasia, villous adenomas, large adenomas >1 cm, or multiple adenomas: three or more) [3,4,5].

The polyamine pathway has been implicated in carcinogenesis, including colorectal tumorigenesis. Germline mutations in the adenomatous polyposis coli (APC) tumor suppressor gene are found in Familial Adenomatous Polyposis (FAP); somatic APC mutations are found in the vast majority of sporadic colorectal cancers. APC regulates ornithine decarboxylase (ODC), which catalyzes polyamine synthesis from ornithine (an arginine-derived product) [6]. Without APC regulation of ODC, polyamine content increases in colorectal tissues, which promotes tumorigenesis [7]. Treatment with the chemopreventive agent L-α-diflouromethylornithine (DFMO, eflornithine), a suicide ODC inhibitor, decreases polyamine levels and suppresses tumorigenesis in the small intestine and colon of FAP-mouse models (Apc^Min/+^ mice) [7]. In a phase III clinical trial of colorectal adenoma patients, polyamine inhibition with combination of low-dose eflornithine (500 mg daily) and sulindac (150 mg daily) resulted in marked inhibition of colonic polyps, regardless of *Odc* genotype [8].

Recent studies have implicated polyamines in the regulation of immune response to immune checkpoint blockade in cancer treatment and demonstrated that polyamine levels are altered by interactions of the microbiome and the tumor microenvironment [9]. Intestinal bacteria can convert arginine directly to polyamines [10]. This route is not essential, however, as the hepatic and extra-hepatic arginases convert arginine to ornithine. Ornithine is itself converted by ODC to form polyamines. Thus, arginine represents an important substrate in the polyamine synthetic pathway. The major source of dietary arginine in humans is meat consumption—particularly beef consumption—but high quantities are found in pork, fish, and chicken [11]. Meat and meat products accounted for 38% of total daily arginine in a study of Finnish men who were estimated to have similar daily arginine intake to U.S. men (i.e., ~5000 mg/day) [10]. Red meat consumption has been associated with an increased risk of colon cancer in humans [12]; however, arginine’s specific contribution to colorectal cancer risk has not been determined. Importantly, arginine is the key substrate for two competing metabolic pathways believed to be involved in carcinogenesis: polyamine synthesis and the nitric oxide (NO) synthase pathway (Figure 1). Nitric Oxide synthase 2 (NOS2) and other NO synthases catabolize arginine to form nitric oxide. Inducible NOS2 isoforms (in addition to cyclooxygenase, COX-2) are abundant in human colorectal adenomas [13]. NOS2 is also overexpressed in rat colon carcinoma tissues after azoxymethane treatment [14,15], an effect that aspirin inhibits [14]. Apc^Min/+^ Nos2 knockout mice provided with dietary arginine showed fewer intestinal adenomas when compared to Apc^Min/+^ Nos2^+/+^ mice [16,17]. Prior experimental and epidemiologic analyses suggest that arginine-restriction [18] and NSAID use [19] confer benefits for colorectal carcinogenesis and survival, particularly among familial CRC patients.

Despite potentially curative treatment, locally advanced colorectal cancer patients remain at considerable risk for multiple CRC-related events, and thus are the target for “tertiary cancer prevention” efforts (i.e., cancer prevention strategies aimed at reducing CRC-related events among patients with prior CRC history). Using our prior results in combination with the available literature, we designed a phase IIa clinical trial among CRC patients in order to detect favorable biomarker alterations in rectal tissue after a 12 week polyamine-inhibitory intervention comprised of dietary arginine restriction and daily aspirin use.

## 2. Methods

### 2.1. Study Design

This was a phase IIa clinical trial with pre- and post-intervention assessments of secondary endpoint biomarkers in colorectal cancer patients (ClinicalTrials.gov Identifier: NCT00578721). As such, all subjects received the intervention. The primary endpoint was to determine if a 12-week intervention consisting of daily enteric coated aspirin 325 mg and adherence to an arginine-restricted diet results in a significant ≥50% decrease in rectal tissue putrescine levels in optimally treated stage I-III colorectal cancer patients. Here, rectal tissue putrescine represents a clinical biomarker. A total of 24 patients were enrolled to ensure that at least 20 completed the study. On-study procedures are shown in Figure 2.

Primary Telephone Counseling (PTC) Randomization Scheme: A subset of eligible participants were chosen at random to receive psychosocial telephone counseling (PTC). PTC was included as an exploratory aim to determine if it could positively reinforce behavioral changes and provide support during the trial. A randomization scheme was constructed for PTC assignment with block size of two (i.e., among 2 consecutive patients, one received PTC and the other did not). A set of consecutively numbered, sealed envelopes each contained a sheet giving the random assignment and spaces to record patient identity. Upon consent, each subsequent envelope was opened, revealing the assignment, and the form completed.

### 2.2. Eligibility Criteria

Eligible patients included those age 18 years or older with a history of completely-resected AJCC stage I, II, or III colon or rectal cancer having no anticipated further treatment with radiation therapy or chemotherapy, and an ECOG Performance Status of 0–1. Patients with FAP or HNPCC were excluded. Patients having already received adjuvant (i.e., post-operative) radiation therapy to the colorectum were excluded. However, patients receiving neoadjuvant radiation therapy with or without chemotherapy prior to surgery for rectal cancer were eligible. Patients were excluded if they had a history of another invasive cancer within 5 years, excepting those with non-melanomatous skin cancer, melanoma in situ, Stage I cervical cancer, or Stage 0 CLL. Patients were required to have normal organ function. Patients already taking aspirin 325 mg/day at baseline were excluded, but those taking 81 mg/day aspirin were allowed on-study, substituting their daily 81 mg aspirin with protocol-defined 325 mg/day aspirin. The protocol did not allow enrolling patients taking calcium supplements (>520 mg/day), corticosteroids, NSAIDs, or anticoagulants on a regular or predictable intermittent basis. Patients with a history of abnormal wound healing or repair were excluded. Patients were excluded if they had personal history of colon resection >40 cm or inflammatory bowel disease. Pregnant or lactating women were not eligible. Premenopausal and perimenopausal women were required to be using adequate birth control methods. No history of allergies or adverse reactions to aspirin was allowed. No documented history of gastric/duodenal ulcer (or active treatment for such) within the last 12 months was allowed.

### 2.3. Intervention and On-Study Assessments

Patients were identified from the outpatient clinics at the University of California Irvine Medical Center (Orange, CA, USA). Upon obtaining written consent, the following clinical procedures were conducted: medical history & physical exam, baseline labs, *Odc* genotyping, three baseline 24 h Food Recall interviews (over a 2 week period) with estimation of total daily meat and arginine intake, plasma biomarker levels, and polyamine contents from the flat rectal mucosa at time of baseline flexible sigmoidoscopy. Participants were asked to complete a 24 h dietary recall interview by telephone with a dietitian approximately 3 separate times over a two week period at baseline, at mid-study (about 6 weeks), and again prior to the end of the study (12 weeks). Dietary intake data were collected and analyzed using Nutrition Data System for Research software version 6.0, developed by the Nutrition Coordinating Center (NCC), University of Minnesota, Minneapolis, MN. The NDS-R software uses the multi-pass system of the interview methodology. Interview prompts guide interviewers and ensure standardized data entry, and dietary interviewers were certified by NCC for use of the NDS-R software. Upon review of dietary intake data, an individualized dietary plan was produced with the research dietician to create a dietary prescription that would reduce estimated arginine intake by 30%. This was performed primarily through lowering meat intake by 50%, and lowering the intake of nuts, certain types of cheese, or other foods high in arginine content for the study duration. The dietitian discussed the diet with each subject to ensure that these dietary restrictions do not result in inadequate amounts of the recommended protein intake for the study period (i.e., 0.8 g protein per kg body weight per day). If the subject was not the person who regularly prepares food at home, the subject was asked to have the household food-preparer present when the dietary prescription was provided. In this manner all patients received an individualized ≥30% arginine-restricted dietary prescription plus aspirin 325 mg PO daily x 12 weeks with nutritional education and compliance assessments. After 12 weeks on-study, patients underwent endoscopy with rectal biopsies for tissue polyamine content, and plasma biomarker levels, end of study H & P, labs, and 3 final 24 h food recall interviews for assessment of total arginine intake at end of study.

Dietary Assessment: Food consumption was assessed via three separate 24 h dietary recall interviews with a study dietician during a 2 week period. These occurred at baseline, at mid-study (about 6 weeks) and again at end-of-study period (12 weeks), for a minimum of 9 dietary assessments over the course of study. Dietary factors that can influence the amount of intestinal polyamines and microbial activity, such as the amount and type of fiber and dietary protein, were also examined for a possible modifying effect on biomarker responsiveness [20].

Plasma Amino acid analysis: Arginine and ornithine concentrations were analyzed as described previously [10], as a measure of biochemical plasma responses to dietary arginine restriction. Plasma arginine correlates directly with arginine intake [10], and plasma ornithine levels can be monitored as a measure of adherence to an arginine-restricted diet [21,22]. Plasma samples were deproteinized with 5% (by volume) sulfosalicylic acid. One part 5% sulfosalicylic acid plus norleucine as an internal standard was added to 2 parts plasma, which was mixed thoroughly and kept at 4 °C for one hour. The samples were centrifuged at 12,000× *g* for 15 min at 4 °C, and the supernatant was used for analysis. Arginine concentrations were analyzed via a Biochrom 20 amino acid analyzer (Pharmacia LKB Biochrom Ltd., Cambridge, UK), which is based on continuous flow ion exchange chromatography with ninhydrin detection of separate amino acids and lithium exchange ion.

Endoscopy with Rectal Mucosal Biopsy Procedure: At study entry (week 2) and at the end of study (week 13), endoscopy was performed and rectal mucosal biopsies were obtained for study evaluation. Eight biopsy specimens were obtained 8–12 cm above the anus and processed for tissue biomarker measurement. Biopsies were snap-frozen in liquid nitrogen and shipped to the University of Arizona for analysis.

Rectal Mucosa Tissue Polyamines: Polyamine content was determined as described previously [23], as this represents an important endpoint of treatment effect on the polyamine pathway. Eight small 4 mm^2^ biopsy specimens were obtained; 3 were randomly selected for polyamine content. These methods measure putrescine, cadaverine, histamine, spermidine, spermine, and monoacetyl derivatives of putrescine, spermidine, and spermine [23]. Quality assurance procedures included regular measurements of standard polyamine preparations and use of internal standards in assessing polyamine amounts. Tumor-free portions of the rectum were collected, flushed with ice-cold saline, and stored frozen at −80 °C. Samples were processed and assayed for polyamine (putrescine, spermidine, spermine, and *N*^1^-acetylspermine) content by reverse-phase high performance liquid chromatography with 1,7-diaminoheptane as an internal standard [24]. Protein content in each sample was determined by the BCA protein assay kit (Pierce, Rockford, IL, USA). Data were expressed as nmol polyamine per mg protein.

### 2.4. Clinical Evaluation

At entry into the study and at 6 and 12 weeks or upon study discontinuation, patients underwent history and physical examination, and blood was drawn for CBC, with a metabolic panel to include liver function tests, BUN, creatinine, and urinalysis. All patients were evaluated for toxicity assessment from the time of first dose of aspirin and dietary arginine restriction (day 1), using Common Terminology for Adverse Events v4.0 (CTCAE, https://evs.nci.nih.gov/ftp1/CTCAE/CTCAE_4.03/Archive/CTCAE_4.02_2009-09-15_QuickReference_8.5x11.pdf) (accessed on 27 February 2023). 

### 2.5. Statistical Analyses

The primary endpoint was percent reduction in rectal mucosa tissue putrescine levels from baseline to end of study. In prior research involving either colorectal adenoma patients [23] or experimental murine models [18], intestinal tissue putrescine reductions after polyamine-inhibitory treatments (DFMO, or celecoxib) ranged from 67–74%. Sample size calculations posited that percent reduction in putrescine would be 30% under the null and 60% under the alternative hypotheses, yielding a requirement for 20 subjects, using a 2-sided test (alpha = 0.05, power = 80%); 24 patients were enrolled to allow for at least 15% attrition. The Sign Rank test was used to compare baseline to end-of-study rectal tissue putrescine levels, as well as levels of spermidine and spermine. All statistical analyses were conducted using SAS 9.1 statistical software (SAS Inc., Cary, NC, USA).

### 2.6. Ethical Considerations & Data Monitoring

This trial was approved by the Institutional Review Board of the University of California, Irvine. After discussing risks and benefits, all willing participants signed a written IRB-approved consent form. Participant safety was monitored by the UC Irvine Chao Family Comprehensive Cancer Center’s Data Safety Monitoring Board (DSMB).

## 3. Results

### 3.1. Demographics

Among 24 eligible patients, 20 patients completed the study, including 10 rectal cancer patients and 10 colon cancer patients. By race/ethnicity, 60% of the patients were White, 40% were Asian, and 5% were Hispanic. 35% of participants were females. Mean age of all participants was 58 years. Patient clinical and pathological demographics are displayed in Table 1.

### 3.2. Dietary Analyses

Macronutrient intake from dietary intakes over the course of the study are presented in Table 2. Total reported kcal intake over the course of the study did not significantly differ pre- (median 1611 kcal/d) or post-intervention (1371 kcal/day, *p* > 0.32). When comparing relevant micronutrient data, the mean reported dietary arginine intake decreased after study intervention from 3.7 g/day ± 1.3 SD to 2.6 g/day ± 1.2 SD (29.7% decrease, *p* < 0.02 by Sign test (Figure 3). There was a significant correlation between the ranks of change in plasma arginine as a function of relative percent change in kCal (Sr = 0.68), with value of relative percent change in Kcal accounting for about 46 percent of the uncertainly in relative percent change of plasma arginine.

### 3.3. Plasma Arginine, Plasma Ornithine Levels

Mean plasma arginine levels decreased significantly from 46.0 ng/mL ± 31.53 SD at baseline to 35 ng/mL ± 21.67 SD at end of study (Mean difference–19.2 ng/mL, Sign Rank *p* < 0.001). Mean ornithine levels, however, did not significantly differ pre- (144.8 ng/mL) vs. post-intervention (133.5 mg/mL, mean difference–6.8 ng/mL, Sign Rank *p* < 0.53).

### 3.4. Rectal Tissue Polyamine Analysis

Rectal tissue polyamine levels were collected and analyzed for all patients; however, paired data were only available for 14 patients due to tissue processing issues affecting 6 patients (polyamine levels could not be calculated from submitted tissues). Rectal tissue putrescine levels (nMol/mg protein) did not differ statistically when comparing pre- (0.90 nMol/mg protein) vs. post- (0.99 nMol/mg protein) intervention biopsies (Sign-rank *p* < 0.64, n = 14, Figure 4). Similarly, no statistically significant differences were observed for rectal tissue changes at end of study vs. baseline for spermidine (Sign rank *p* < 0.13), spermine (Sign rank *p* < 0.21), or the spermidine to spermine ratio, SSR (Sign rank *p* < 0.71).

### 3.5. Adverse Events

Among the 20 patients completing the study, there were 7 therapy-related adverse events, all of which were grade 1 (diarrhea n = 2, fatigue n = 3, neuropathy n = 1, and myalgia n = 1). Of note, no clinically significant rectal bleeding occurred in any patient at baseline or end-of-study biopsies.

### 3.6. Quality of Life/Psychosocial Telephone Counseling (PTC) Intervention

Eight (8) patients were randomized to receive PTC. Of patients randomized to PTC, 71.4% achieved a lower dietary arginine consumption than the median decline, compared to 40% of patients randomized not to have PTC (*p* < 0.34, NS). Seven of eight rated each of the counseling sessions as “quite a bit” to “very” useful (highest score possible). In addition, seven of eight patients rated the counseling programs as “very helpful” (highest score possible). In response to the question, “Please indicate if any of the following changes occurred as a result of participating in the counseling sessions”, the majority checked a mean of five different changes (Range: 2–9 changes). Prominent changes included: improved diet, increased frequency of exercise, increased motivation for making healthy lifestyle improvements, and improved stress management.

## 4. Discussion

In this phase IIa tertiary prevention clinical trial of CRC survivors, an individualized dietary prescription to reduce dietary arginine intake by ≥30% along with daily intake of 325 mg oral aspirin did not significantly reduce rectal tissue polyamine levels. As such, the polyamine-inhibitory intervention failed to demonstrate a biomarker effect in the target tissue of origin (i.e., rectal mucosa); this is one of the key criteria for further cancer prevention clinical trial development established in the landmark AACR consensus statement [25]. Of note, our trial demonstrated that this novel approach was safe and feasible, as we successfully achieved the 30% dietary arginine restriction goal among trial participants without substantial adverse effects, resulting in statistically significant decreased plasma arginine levels. As such, the lack of tissue biomarker response indicates that potential rectal tissue biomarker effects of 12 weeks of combination dietary arginine restriction and aspirin do not recapitulate the pharmacologic effects of polyamine inhibition through eflornithine and/or NSAIDs, as demonstrated in other studies.

Lowering rectal mucosal polyamines predicts pharmacologic treatment’s ability to prevent colorectal adenomas [26]. However, as mentioned, we did not observe a tissue polyamine-lowering effect of our dietary arginine restriction and aspirin intervention. It is possible that the 12 week intervention duration was too short for our intervention to result in 50% tissue putrescine reduction. This treatment duration was selected based on results from two separate trials involving tissue polyamine reductions after polyamine-inhibitory treatment with DFMO. In a prior colon cancer chemoprevention trial, treatment with oral DFMO at 0.4 g/m^2^ resulted in a reduction of the rectal mucosa tissue spermidine:spermine ratio at 6 months that was similar to the reduction noted at 12 months [23]. In a small, phase IIa trial of nine prostate cancer patients, a short 4 week course of oral DFMO at 0.5 g/m^2^ resulted in significant reductions of prostatic tissue polyamine content [27]. Thus, the selected duration here was intermediate between 1 and 6 months as noted in the above polyamine-inhibition trials using DFMO, and theoretically sufficient to observe the proposed changes in tissue polyamines pre- vs. post-intervention.

Aspirin and non-steroidal anti-inflammatory drugs (NSAIDs) inhibit intestinal cancer development in mouse models [18,28]. Clinical trials have implicated aspirin in premalignant colorectal adenoma regression and prevention [29,30,31], and cohort and case-control studies have shown that along with other NSAIDs, aspirin reduces CRC risk [32,33,34,35], particularly with prolonged use [34]. Lack of consensus, and unexpected cardiovascular toxicities associated with celecoxib use preclude recommending NSAIDs for primary prevention of CRC [36,37]. In a change from the 2016 US Preventive Services Task Force (USPSTF) guidelines, new 2022 guidelines state there is currently insufficient evidence to support aspirin to reduce incidence or mortality of CRC [38].

A novel aspect of this phase IIa clinical trial is the Psychosocial Telephone Counseling (PTC) intervention. Psychosocial Telephone Counseling has shown benefit among cancer survivor cohorts in other studies, particularly in effects on overall quality of life, social support, and survivor-specific issues [39,40]. Interestingly, such PTC-based changes are marked by beneficial changes in patients’ blood-based immune profile [39,40]. To our knowledge, these potentially efficacious PTC interventions have not been previously conducted among CRC survivors. While our numbers are small, seven of eight patients randomized to the PTC intervention reported that it was “very helpful” (highest score), and the majority indicated that PTC helped to improve diet, increase frequency of exercise, increase motivation for making healthy lifestyle improvements, and improve stress management. In addition, while not statistically significant, it is interesting to observe that patients randomized to PTC had numerically greater reductions in reported dietary arginine intake over the 12 week intervention period. Dietary interventions present unique challenges in the clinical trial setting. Given our encouraging but preliminary results, PTC may have a role in facilitating adherence to dietary treatments or other biobehavioral interventions in the clinical trial setting.

While adequately powered to address the study question, the study is limited by small sample size and was not powered to identify subtle polyamine tissue biomarker effects. In addition, the small sample size precludes one from making meaningful assessments of any potential differential effects by *Odc* genotype, which have been implicated as relevant in a larger clinical trial [41] and in two epidemiologic cohort studies—one of which demonstrated differential effects of meat consumption on outcomes based on *Odc* genotype [42,43]. By reducing dietary arginine, our study focused on reducing endogenous polyamine synthesis. Others have investigated effects of exogenous polyamine intake on CRC risk. In a study of individuals in China [44], differential effects based on specific dietary polyamine intake were reported (beneficial for putrescine and spermidine, adverse risk for spermine). In a separate analysis of data involving U.S. women enrolled in the Women’s Health Initiative cohort study, no positive associations were observed for dietary polyamine intake and CRC risk, or CRC-specific mortality [45].

In our phase IIa clinical biomarker trial, patients were heterogeneous with respect to primary tumor site (colon, rectum), stage, and duration of remission since time of initial operation, which may have confounded the analysis. While the study (and intervention itself) relied on numerous dietary food recalls under the direction of a research dietitian, the nature of this methodology is dependent on patient recall and thus may be inaccurate.

Despite the negative rectal tissue biomarker results of this dietary and aspirin-based polyamine-inhibitory intervention, secondary and tertiary prevention of CRC through pharmacologic polyamine inhibition remain active areas of investigation. To this end, the randomized clinical trial of eflornithine, sulindac, and placebos in FAP patients (secondary prevention trial) provides some insights [46]. In the study, FAP patients randomized to one of three study arms (eflornithine + sulindac vs. eflornithine + sulindac-placebo vs. eflornithine-placebo + sulindac) revealed no significant differences on disease progression. However, among patients with an intact colorectum (“pre-colectomy group”), interesting yet non-significant differences were reported: time to disease progression was 39 months in the eflornithine-sulindac group, 25 months in the sulindac group, and 20 months in the eflornithine group. In a post-hoc analysis of this FAP trial, combination eflornithine-sulindac was shown to delay or prevent the need for lower gastrointestinal surgery in FAP patients [47]. Two additional U.S. National Cancer Institute (NCI)-supported phase III clinical trials are relevant to understanding pharmacologic polyamine-inhibitory agents as tertiary CRC prevention. Results of the large phase III clinical trial results of C80702 (ClinicalTrials.gov Identifier: NCT01150045), which randomized stage III colon cancer patients to a 3 year intervention of celecoxib or placebo after 1st randomization to 3 vs. 6 mo adjuvant FOLFOX chemotherapy, revealed no difference in disease-free survival ascribed to celecoxib [48]. In a follow-up analysis of patients in this trial based on plasma inflammatory markers taken 3–8 weeks post-operatively (and prior to chemotherapy), higher inflammation postoperatively was associated with increased risk of recurrence or death [49]. Of note, analyses of the secondary clinical endpoints for C80702 (e.g., total adenoma recurrence, high-risk adenoma recurrence) have not yet been reported. Currently, accrual is nearing completion for the NCI-National Community Oncology Research Program (NCORP)-sponsored trial SWOG 0820: a double-blind randomized controlled clinical trial of eflornithine and sulindac vs. placebos in stage 0-III CRC patients (ClinicalTrials.gov Identifier: NCT01349881) [50]. The primary endpoint of S0820 is to determine if there is a ≥60% reduction in high-risk adenomas and 2nd primary CRCs after 3 year intervention of oral eflornithine + sulindac vs. matching placebos in resected CRC patients.

In conclusion, our phase IIa clinical trial of a 12 week intervention of daily oral aspirin 325 mg plus an individualized dietary arginine restriction regimen did not result in rectal tissue polyamine changes in CRC survivors. Whether a more stringent dietary arginine restriction prescription, or a prolonged duration of intervention would have yielded different results is unknown and cannot be imputed from this study. Nevertheless, there is a disconnect between our results here and the tissue polyamine-lowering effects of various therapeutic prevention agents such as eflornithine, sulindac, aspirin, and various NSAIDs. As such, we await the results of the primary and secondary analyses of key NCI-sponsored CRC therapeutic prevention clinical trials, as well as ongoing basic and translational research that will shed light on potential future roles for polyamine inhibition as CRC prevention.

## Figures and Tables

**Figure 1 cancers-15-02103-f001:**
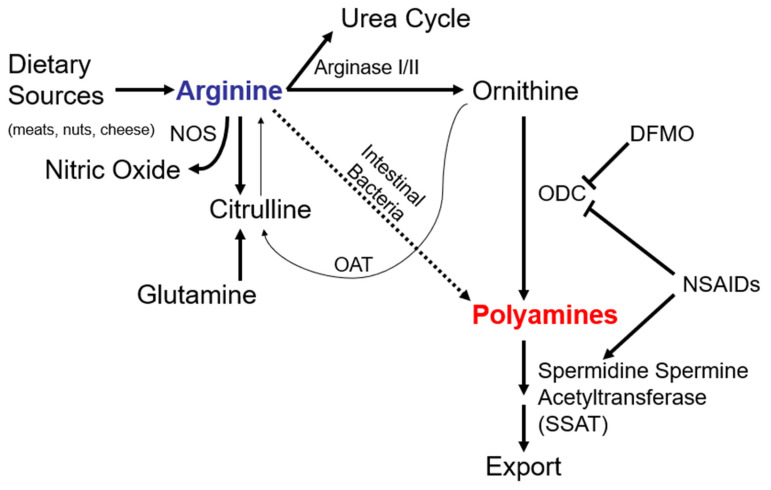
Arginine as the central substrate for polyamine and nitric oxide metabolism. Alternate pathways and inhibitors are indicated. NOS, nitric oxide synthase; DFMO, difluoromethylornithine (eflornithine); ODC, ornithine decarboxylase; NSAIDs, non-steroidal anti-inflammatory drugs; SSAT, spermidine/spermine *N*1-acetyltransferase; OAT, ornithine aminotransferase.

**Figure 2 cancers-15-02103-f002:**
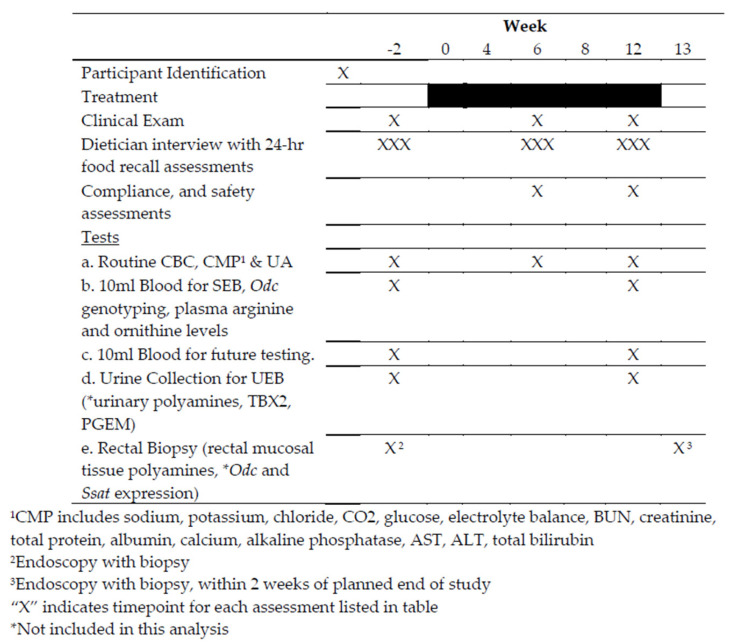
Clinical procedures involved in the Phase IIa Clinical Trial.

**Figure 3 cancers-15-02103-f003:**
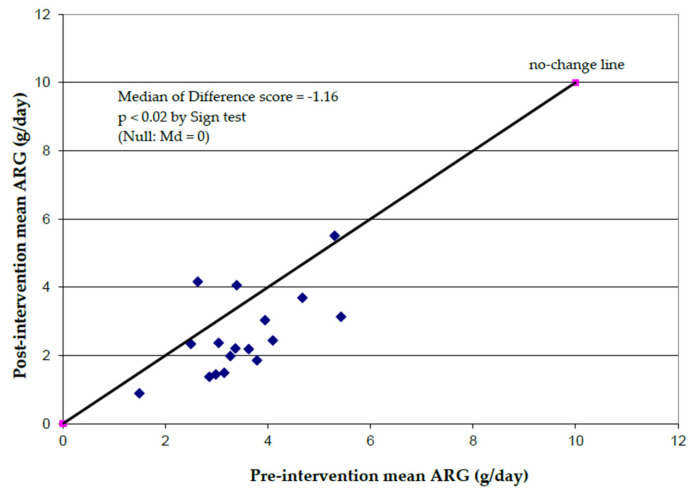
Dietary Arginine Intake, post- vs. pre-intervention (dietary arginine restriction and oral aspirin 325 mg daily). Measurements missing for n = 3 patients.

**Figure 4 cancers-15-02103-f004:**
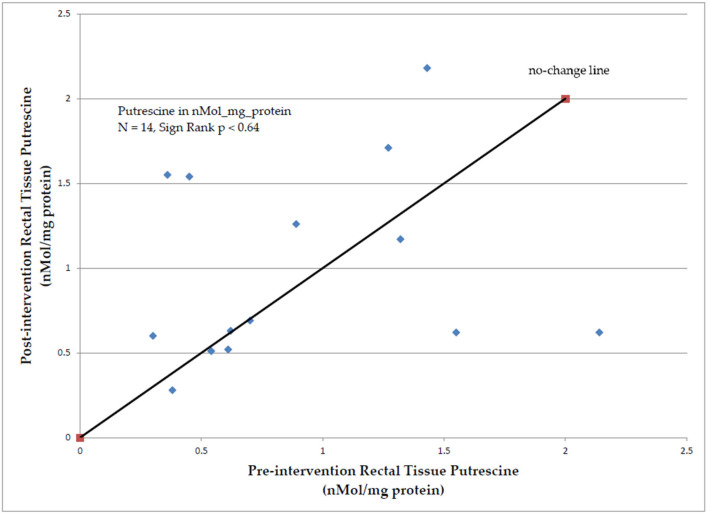
Rectal Tissue Putrescine Levels, post- vs. pre-intervention (dietary arginine restriction and oral aspirin 325 mg daily).

**Table 1 cancers-15-02103-t001:** Clinical and Demographic Characteristics. PTC, Psychosocial Telephone Counseling.

		Total (n = 20)
Gender	Male	13 (65%)
	Female	7 (35%)
Mean Age	Male	62.1 y (12.3)
	Female	51.7 y (6.9)
	Total	58.1 y (11.6)
Race	Asian	8 (40%)
	White	12 (60%)
Ethnicity	Hispanic	1 (5%)
	White	19 (95%)
Tumor Subsite Location	Cecum	1 (4.8%)
	Transverse	1 (4.8%)
	Sigmoid	5 (23.8%)
	Rectosigmoid	4 (19%)
	Rectum	6 (28.6%)
	Colorectum-NOS	3 (15%)
AJCC Stage	I	4 (19%)
	IIA	4 (19%)
	IIIA	3 (14.3%)
	IIIB	8 (38.1%)
	IIIC	1 (4.8%)
Tumor Grade	Well-Differentiated (Gr 1)	3 (15%)
	Moderately Differentiated (Gr 2)	10 (50%)
	Poorly Differentiated (Gr 3)	1 (5%)
	Missing	6 (30%)
*Odc* Genotype	AA	4 (20%)
	AG	7 (35%)
	GG	9 (45%)
Randomized to PTC Intervention?	Yes No	8 (40%) 7 (35%)
	N/A	5 (25%)

**Table 2 cancers-15-02103-t002:** Macronutrient data calculated from patient reported 24-h food recall surveys: minimum of 3 surveys done per patient at each timepoint: Baseline, Midpoint (6 weeks on-study), and End of study (12 weeks).

Variable	N	Mean	Median	Std Dev	Minimum	Maximum
Carbohydrate (g) Baseline Midpoint End of Study	20 17 17	216.5 210.1 189.7	200.4 193.9 182.5	101.4 105.9 58.0	97.2 74.9 75.0	518.9 563.9 298.1
Fiber (g) Baseline Midpoint End of Study	20 17 17	18.0 18.5 17.4	18.4 16.5 16.1	7.3 10.5 6.4	6.6 5.6 7.0	31.8 48.2 33.1
Fat (g) Baseline Midpoint End of Study	20 17 17	56.5 51.4 49.2	49.8 52.4 48.2	29.6 32.9 19.0	16.7 17.6 14.8	160.4 154.6 79.8
Protein (g) Baseline Midpoint End of Study	20 17 17	65.4 55.9 53.5	62.9 54.3 49.4	20.9 20.5 20.3	27.8 27.8 16.7	118.7 112.4 92.7
Energy (kcal) Baseline Midpoint End of Study	20 17 17	1625.5 1515.7 1409.8	1595.1 1388.4 1445.1	682.2 766.7 401.9	661.4 630.0 508.4	3925.5 4090.4 2255.3

## Data Availability

The data generated in this article will be shared upon reasonable request to the corresponding author.

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
