# Peer review of "Phase IIa Clinical Biomarker Trial of Dietary Arginine Restriction and Aspirin in Colorectal Cancer Patients"

_cancers, 2023, doi:10.3390/cancers15072103_

Round 1

Reviewer 1 Report

In general, this article is very suitable for the design of the research and the expression of the content, and it is okay from the report of the results of the research stage, and it is better to simplify and understand the language expression. In addition, it was suggested that a complete concluding paragraph be added.

Author Response

Author’s Response to R.1: Thank you for your constructive feedback. In response, we have simplified the text of the manuscript for clarity (see detailed response to Reviewer 3 queries), and have added a Conclusion to the Discussion (as also requested by Reviewer 3). 

Reviewer 2 Report

The purpose of this research study is well presented. 

Author Response

Author’s Response to R.2: Thank you for your constructive feedback. 

Reviewer 3 Report

Comments :

1.       Aspirin is one of those anti-inflammatory drugs that thin the blood and decrease blood clotting. How the patient behaves after rectal biopsies collections?

2.       The manuscript is not well-presented in the template.

3.       In the introduction the authors said that (…Globally, CRC ranks as the 3nd most common form of cancer with 1.9 million cases in 2020..). Does the word (globally) means (worldwide)?

4.       In this same sentence, (3nd) should 3rd or 2nd.

5.       In the introduction: this sentence (Aside from risk of recurrence and CRC-specific mortality, CRC patients remain at high risk for multiple competing events, including a 50% risk of recurrent adenoma and 27% risk of either 2nd primary colorectal cancer or high-risk adenoma (defined as adenomas with high-grade dysplasia, villous adenomas, large adenomas >1cm, or multiple adenomas defined as 3 or more) [3-5].) is too complicated. The reviewer suggests splitting it into at least two different sentences.

6.       The authors said (…Preliminary experimental and epidemiologic analyses suggest that arginine-restriction [18] and NSAID use [19] confer benefits for colorectal carcinogenesis and survival, particularly among familial CRC patients.). this reviewer suggests changing (Preliminary) to (previous) as the results were already published.

7.       Those patients taking 81mg/day aspirin were allowed on-study, do they have additional aspirin to reach the 325mg/day to get involved in the study?

8.       One of the exclusion criteria was (.. Patients with a history of abnormal wound healing or repair were excluded.). Is this referring to diabetic patients?

9.       In the last sentence of the abstract, the authors said (…Among CRC survivors, treatment with daily oral aspirin and dietary arginine restriction resulted in lower dietary arginine intake and plasma arginine levels but did not affect rectal tissue polyamine levels.). It is obvious that, a dietary restriction in arginine leads to a decreasing arginine intake and plasma levels. Furthermore, in the title, it states (clinical biomarker). What this clinical biomarker refers to in this study? 

10.   There is no conclusion at the end of the discussion section.

This study could be of some interest to the field and the readers. It could be accepted for publication in this journal.

Author Response

Reviewer 3
1.    Aspirin is one of those anti-inflammatory drugs that thin the blood and decrease blood clotting. How the patient behaves after rectal biopsies collections?

Authors’ Response: No clinically significant rectal bleeding occurred after rectal biopsy among the 20 patients on-study at pre-intervention, or post-intervention. This information is added to the Results section of the revised manuscript (Section 3.5, Adverse Events):
“Of note, no clinically significant rectal bleeding occurred in any patient at baseline or end-of-study biopsies.”

2.    The manuscript is not well-presented in the template.

Authors’ Response: We believe the new Cancers/MDPI format greatly facilitates the reader in terms of presentation and appreciate the editor’s template revisions. 

3.    In the introduction the authors said that (…Globally, CRC ranks as the 3nd most common form of cancer with 1.9 million cases in 2020..). Does the word (globally) means (worldwide)?

Authors’ Response: Correct, and to clarify we have replaced “Globally” with “Worldwide” in the revised Introduction section. 

4.    In this same sentence, (3nd) should 3rd or 2nd.

Authors’ Response: thank you for noticing this typo. The Introduction of the revised manuscript now clearly states “3rd” most common form of cancer.

5.    In the introduction: this sentence (Aside from risk of recurrence and CRC-specific mortality, CRC patients remain at high risk for multiple competing events, including a 50% risk of recurrent adenoma and 27% risk of either 2nd primary colorectal cancer or high-risk adenoma (defined as adenomas with high-grade dysplasia, villous adenomas, large adenomas >1cm, or multiple adenomas defined as 3 or more) [3-5].) is too complicated. The reviewer suggests splitting it into at least two different sentences.

Authors’ Response: Thank you. We have split this sentence into 2 separate sentences, as the reviewer recommends, to provide clarity, as follows: 
“Aside from risk of recurrence and CRC-specific mortality, CRC patients remain at high risk for multiple competing events.  Such events include a 50% risk of recurrent adenoma and 27% risk of either 2nd primary colorectal cancer or high-risk adenoma (i.e., high-grade dysplasia, villous adenomas, large adenomas >1cm, or multiple adenomas: 3 or more) [3-5].

6.    The authors said (…Preliminary experimental and epidemiologic analyses suggest that arginine-restriction [18] and NSAID use [19] confer benefits for colorectal carcinogenesis and survival, particularly among familial CRC patients.). this reviewer suggests changing (Preliminary) to (previous) as the results were already published.

Authors’ Response: we appreciate the comment and have revised the text accordingly for clarity, as follows: 

“Previous experimental and epidemiologic analyses suggest that arginine-restriction [18] and NSAID use [19] confer benefits for colorectal carcinogenesis and survival, particularly among familial CRC patients. Despite potentially curative treatment, locally-advanced colorectal cancer patients remain at considerable risk for multiple CRC-related events, and thus are the target for “tertiary cancer prevention” efforts (ie, cancer prevention strategies aimed at reducing CRC-related events among patients with prior CRC history). Given our prior results in combination with the available literature…”

7.    Those patients taking 81mg/day aspirin were allowed on-study, do they have additional aspirin to reach the 325mg/day to get involved in the study?

Authors’ Response: An excellent point- we have clarified this in the text of the revised manuscript with the following statement:
“Patients already taking aspirin 325mg/day at baseline were excluded, but those taking 81mg/day aspirin were allowed on-study, substituting their daily 81mg aspirin with protocol-defined 325mg/day aspirin” 

8.    One of the exclusion criteria was (.. Patients with a history of abnormal wound healing or repair were excluded.). Is this referring to diabetic patients?

Authors’ Response:  No, this statement is not limited to diabetic patients. Rather, anyone with history of abnormal wound healing or repair was excluded, as stated. 

9.    In the last sentence of the abstract, the authors said (…Among CRC survivors, treatment with daily oral aspirin and dietary arginine restriction resulted in lower dietary arginine intake and plasma arginine levels but did not affect rectal tissue polyamine levels.). It is obvious that, a dietary restriction in arginine leads to a decreasing arginine intake and plasma levels. Furthermore, in the title, it states (clinical biomarker). What this clinical biomarker refers to in this study? 

Authors’ Response: To clarify, we have revised the last sentence of the abstract as follows:
“Among CRC survivors, treatment with daily oral aspirin and an individualized dietary arginine restriction intervention resulted in lower calculated dietary arginine intake and plasma arginine levels but did not affect rectal tissue polyamine levels.”
The clinical biomarker is rectal tissue putrescine levels pre vs. post- intervention (ie, our primary endpoint). In order to clarify, we have added text to the Methods section as follows:
“The primary endpoint was to determine if a 12-week intervention consisting of daily aspirin 325mg and adherence to an arginine-restricted diet results in a significant > 50% decrease in rectal tissue putrescine levels in optimally-treated stage I-III colorectal cancer patients. Here, rectal tissue putrescine represents a clinical biomarker.”

10.    There is no conclusion at the end of the discussion section.

Authors’ Response: We have added the following concluding paragraph in response to this reviewer comment (and that of Rev. 1 above):
“In conclusion, our phase IIa clinical trial of a 12-week intervention of daily oral aspirin 325mg plus an individualized dietary arginine restriction regimen did not result in rectal tissue polyamine changes in CRC survivors. Whether a more stringent dietary arginine restriction prescription, or a prolonged duration of intervention would have yielded different results is unknown and cannot be imputed from this study. Nevertheless, there is a disconnect between our results here and the tissue polyamine-lowering effects of various therapeutic prevention agents such as eflornithine, sulindac, aspirin, and various NSAIDs.  As such, we await the results of the primary and secondary analyses of key NCI-sponsored CRC therapeutic prevention clinical trials, as well as ongoing basic and translational research that will shed light on potential future roles for polyamine inhibition as CRC prevention. ”

Reviewer 4 Report

First of all, thank you so much for involving me in reviewing this manuscript.

Very interesting and topical topic always of great study and debate.

Complex but well-conducted and understandable statistical analysis with well-structured graphs.

Clear and easily understood English language.

Adequate and recent bibliography.

Clear and understandable tables and images.

For me the article is fine.

Author Response

Author’s Response to R.4: Thank you for your constructive feedback.